# Personality Disorders as a Possible Moderator of the Effects of Relational Interventions in Short-Term Psychoanalytic Psychotherapy with Depressed Adolescents

**DOI:** 10.3390/ijerph191710952

**Published:** 2022-09-02

**Authors:** Hans Ole Korsgaard, Randi Ulberg, Benjamin Hummelen, Nick Midgley, Agneta Thorén, Hanne-Sofie Johnsen Dahl

**Affiliations:** 1The Nic Waal Institute, Lovisenberg Hospital, 0440 Oslo, Norway; 2Institute of Clinical Medicine, University of Oslo, 0316 Oslo, Norway; 3Department of Psychiatry, Diakonhjemmet Hospital, 0370 Oslo, Norway; 4Research Unit, Division of Mental Health, Vestfold Hospital Trust, 3116 Tønsberg, Norway; 5Division of Mental Health and Addiction, Oslo University Hospital, 0424 Oslo, Norway; 6Research Department of Clinical, Educational and Health Psychology, University College London (UCL), London WC1E 6BT, UK; 7Anna Freud National Centre for Children and Families, London N1 9JH, UK; 8The Erica Foundation, 114 24 Stockholm, Sweden; 9Department of Psychology, University of Oslo, 0316 Oslo, Norway

**Keywords:** psychodynamic, psychoanalytic psychotherapy, transference, adolescent, depression, personality disorder

## Abstract

A significant proportion of adolescents suffering from major depressive disorder (MDD) are likely to have a co-morbid personality disorder (PD). Short-term psychoanalytic psychotherapy (STPP) was found to be one treatment of choice for adolescents suffering from MDD. Background: The first experimental study of transference work-in teenagers (FEST-IT) demonstrated the efficaciousness of transference work in STPP with adolescents suffering from MDD. The usefulness of STPP may be enhanced by exploring possible moderators. Methods: Depressed adolescents (*N* = 69), aged 16–18 years, were diagnosed with the structured interview for DSM-IV PDs and randomized to 28 weeks of STPP with or without transference work. A mixed linear model was applied. The moderator effect was investigated by a three-way interaction including “time”, “treatment group” and “number of PD criteria”. Results: A small but significant moderator effect was found for cluster B personality pathology. Patients with a higher number of cluster B PD criteria at baseline did better up to one-year post-treatment where therapists encouraged patients to explore the patient–therapist relationship in the here and now. Conclusion: When treated with psychoanalytic psychotherapy for MDD, adolescents with cluster B PD symptoms seem to profit more from transference work than adolescents without such pathology.

## 1. Introduction

Adolescence is a developmental period marked by significant biological, psychological and social changes [1,2]. The exploration and reformation of identity and interpersonal relationships are core developmental tasks for the young person, often affecting their emotional stability. For some individuals, this period means dealing with more severe mood regulation disorders, such as depression, causing persistent feelings of despair and loss of interest in activities.

Depression in adolescence is one of the leading causes of illness and disability in this age group and the prevalence is rising [3,4]. Depression is more frequent in young girls than in boys [5]. Especially when left untreated, it may have severe prognostic consequences reaching far into adulthood [6,7]. Not only are psychiatric disorders such as depression causing widespread mental suffering, the rate of vocational disability due to mental health problems in European young adults has currently risen to as much as 25% [8]. Early loss and relational trauma predispose the development of youth depression and have negative effects on the person’s relationships [1]. The diagnostic criteria for major depressive disorder (MDD) in adolescents and adults are similar. Additionally, the diagnostic criteria are the same in DSM-IV [9] and DSM-5 [10] in young people. The symptoms in adolescent MDD are sadness, irritability, loss of interest or loss of pleasure, appetite and/or sleep disturbances, loss of energy, low self-esteem, reduced concentration, social withdrawal and a sense of hopelessness. In severe cases, there may also be a strong sense of guilt and suicidal ideation.

Adolescence appears also to be a particularly vulnerable period for the development of more complex psychiatric disorders, such as PDs [8]. Borderline personality disorder (BPD) is characterized by severe dysregulation in mood, interpersonal relationships, identity and behaviors. The symptoms may seem consistent with the typical adolescence features but can be clearly distinguished from these by its severe, pervasive and persistent nature. Although often examined in isolation, major depressive disorder (MDD) shows a high level of co-morbidity with a range of PDs, especially BPD [11].

In both DSM-5 and ICD-11, personality disorders (PD) are defined as relatively enduring and maladaptive patterns of experiencing life, coping with problems and relating to others. To diagnose a PD, it is a prerequisite condition that appropriate symptoms are clearly manifested before the age of 18 [10,12]. Still, there has been great reluctance among clinicians to diagnose PDs in adolescents. Concerns about the similarities with normal developmental traits in adolescence, that the personality is not yet consolidated at this stage, as well as the stigma that a PD diagnosis might bring to a young person’s identity, have been claimed as arguments against PD diagnoses in the teenage years [13]. In recent years, however, the concept of PD in adolescence and the prognostic significance of early intervention has gained increasing academic as well as clinical attention [13,14,15,16,17,18,19].

Earlier studies indicated that as many as 25% of adolescents treated in mental health outpatient clinics meet the diagnostic criteria for borderline personality disorder (BPD) alone, rising to 49% in inpatient units [13,15,20]. A study of unselected adolescents referred to a non-specialized mental health outpatient clinic showed that 22% of the referred adolescents had at least one PD, with cluster B (i.e., mainly borderline) and cluster C (i.e., mainly avoidant and dependent) being the most prevalent PDs [11]. Hence, the distribution of PDs among adolescents did not differ from the distribution found in equivalent adult samples. To further emphasize the similarities in clinical appearance between adolescent and adult PDs, the adolescents displayed the same linear relationship between symptom severity and quality of life as their adult counterparts [11].

Timely diagnosis of PDs in adolescents has been shown to be important in preventing the emergence of prolonged co-morbid conditions, such as substance use disorders, anxiety and depression [14,15,16,17,18,19,21,22,23,24,25], as well as reliance on public welfare assistance for support in adulthood [24]. When assessing direct medical costs and indirect losses in productivity, the combined economic burden of PDs, in particular BPD, exceeds by far those of common symptom disorders such as depression or anxiety [23,24,26].

In addition to early detection, the use of therapeutic interventions specifically addressing the interpersonal functioning have been shown in adult studies to be helpful in changing the course of the disorder [27]. However, the number of studies on the treatment of adolescent PDs is relatively sparse and mostly linked to niche programs in specialist services for selected patients [28]. More accessible treatment models are currently being explored, i.e., ‘good psychiatric management’, which involves less intensive, easier-to-learn therapies that have been shown to be nearly as effective as more developed approaches, such as dialectical behavior therapy (DBT) and mentalization-based therapy (MBT) [29]. Notwithstanding the choice of treatment model, personality traits influence the effect of psychotherapy [30].

Several specialized models have been developed for the treatment of specific PD symptoms in adults [31,32,33,34]. MBT was developed as treatment for BPD with the aim of targeting the mentalization deficits thought to be rooted in early attachment insecurity [31]. Kvarstein and colleagues report results indicating that MBT may also apply for BPD patients with severe conditions [35].

In psychodynamic psychotherapy, the aim is to increase awareness of maladaptive patterns in relating to other people. Transference is a basic concept in psychodynamic theory [36], and several psychodynamics-based therapies utilize this concept to varying degrees [31,37,38]. Transference may be defined as representations of important figures from one’s past and the feelings associated with those figures that shape the patient’s perception and interpretation of relational experiences in therapy, leading to somewhat stereotyped or maladaptive emotional responses [39]. Transference work, i.e., the analysis of transference, is considered a fundamental technique in psychodynamic psychotherapy with adults as well as adolescents. It maintains a focus on themes and conflicts that arise in the therapeutic relationship, as opposed to a non-transference focus where the interaction between patient and therapist will not be specifically targeted; without transference work, the focus of treatment will be upon problems in the patient’s relationships outside therapy [27,40,41]. By gaining a better understanding of these patterns, especially when explored ‘live’ in the relationship to the therapist, it is assumed that the quality of social relationships will be improved, which in turn may contribute to better self-esteem and a decrease in depressive symptoms. Thus, transference work maintains a focus on themes and conflicts that arise in the therapeutic relationship, as opposed to a non-transference focus where the interaction between patient and therapist will not be specifically targeted; instead, focus will be upon problems in the patient’s relationships outside therapy [40,41,42].

Kernberg developed an empirically supported model for how transference-based interventions can be helpful in the treatment of personality disorders [42]. Høglend and colleagues reported that patients with PD (cluster C and milder cluster B) improved more and with longer-lasting effects after psychodynamic psychotherapy (PDT) with transference interventions (TI) versus PDT without TI [43]. However, the impact of difficulties in relational functioning and/or personality disorder on the effect of psychodynamic psychotherapy in adolescents has not yet been empirically explored [31].

From a clinical perspective, it might be expected that patients with significant interpersonal problems may especially profit from transference interventions since this work may correct misconceptions about the therapist and increase trust towards them, which in turn will build a good therapeutic alliance—a prerequisite for therapeutic change. Borderline personality disorder is the quintessential mental disorder characterized by interpersonal problems and self-pathology, including identity problems, unstable relationships and emotional dysregulation. In DSM-IV, BPD is situated within cluster B, together with antisocial PD, narcissistic PD and histrionic PD. Though the cluster classification was discontinued in DSM-5, cluster B PDs may still have clinical utility, not least because it aligns well with the externalizing spectrum disorders as outlined by the HitOP model [44], an alternative classification system of mental disorders. Cluster C PDs (avoidant PD, obsessive-compulsive PD and dependent PD), on the other hand, can be assumed to represent the internalizing spectrum whereas cluster A PDs (paranoid, schizotypal and schizoid PD) may be considered as belonging to the perceptual dysregulation dimension. It might be of considerable clinical interest to examine whether the different PD clusters respond differentially to transference interventions.

Short-term psychoanalytic psychotherapy (STPP) is an evidence-based approach to the treatment of depression in adolescence, which aims to help the adolescent achieve a healthy social development, with peers and parents also preparing for the transition from adolescence to participation in adult life [45]. STPP engages the young person in the search for an understanding of and an insight into their own relationships, feelings and the background for the choices they make. The therapist’s role is to help the young person to understand more of the repetitive patterns that may be happening outside the young person’s awareness. “This attentiveness to unconscious phenomena is specific to psychoanalytic psychotherapy, and is related to the theoretical importance attributed to these deeper, less accessible layers of the mind.” [46]. To help improve dynamic change in interpersonal capacities, the therapists in dynamic/psychoanalytic psychotherapy make use of transference work (TW). TW is thought to be a key ingredient in this kind of psychotherapy.

The first experimental study of transference work-in teenagers (FEST-IT) is a multicenter observer- and patient-blind, randomized controlled component study that compares the effects of psychoanalytic psychotherapy with and without transference work in the treatment of MDD in adolescents. The study was testing a model of STPP that had already demonstrated its effectiveness in a previous RCT study [45]. In the FEST-IT study, the therapists were instructed to use (or withhold) five categories of TWs [43,47]:(1)The therapist addressed transactions in the patient–therapist relationship.(2)The therapist encouraged exploration of thoughts and feelings about the therapy and the therapist’s behavior.(3)The therapist encouraged patients to discuss how they believed the therapist might feel or think about them.(4)The therapist included him-/herself explicitly in interpretive linking of dynamic elements (conflicts), direct manifestations of transference and allusions to the transference.(5)The therapist interpreted repetitive interpersonal patterns (including generic interpretations) and linked these patterns to transactions between the patient and the therapist.

The FEST-IT study [48] demonstrated the efficaciousness of transference work on the level of depressive symptoms in psychoanalytic psychotherapy within a group of adolescents suffering from MDD. The clinical usefulness of this finding, however, could be further enhanced by exploring possible moderators, such as the presence of PDs, thereby facilitating a more tailored treatment approach to depressed adolescents. The aim of the present study was, according to the second protocol analysis, to analyze the moderating effect of personality disorders on treatment outcomes in depressed adolescents who received STPP with or without transference work.

## 2. Materials and Methods

### 2.1. Study Design and Participants

The design of the first experimental study of transference work-in teenagers (FEST-IT) is elaborated in detail in Ulberg et al., 2012 [38] and Ulberg et al., 2021 [48].

Data from 69 patients were included in the intention-to-treat analyses. The participating adolescents (*N* = 69), aged 16–18 years, were subjected to an extensive diagnostic assessment procedure, including the structured interview for DSM-IV personality disorders (SIDP-IV) [47]. Then, participants were randomized to 28 weeks of psychoanalytic psychotherapy with (*N* = 39) or without (*N* = 31) transference work. Comorbidity was expected to be frequent in this clinical sample. Symptom diagnoses (Axis I) and personality diagnoses (Axis II) were based on the Mini International Neuropsychiatric Interview (M.I.N.I.) [49] and SIDP-IV, respectively [47,50] (Table 1).

The primary outcome as measured by the Psychodynamic Functioning Scale (PFS; measuring quality of relations with family and friends, tolerance for affects, insight and problem-solving capacity) [51], did not significantly differ between the groups at pre-, post- and one-year post-treatment. However, depression measured with the Beck Depression Inventory (BDI) [52] and Montgomery and Åsberg Depression Rating Scale (MADRS) [53], showed significantly better outcomes from 12 weeks in treatment to the one-year follow-up in the transference work group [38,48].

The randomization was stratified and blinded for the patients and the evaluators. For each of the four patients randomized to each therapist, two patients were treated with and two were treated without transference work.

All therapists treated patients in both treatment groups. The treatment manual developed by the IMPACT research group, and used in the short-term psychoanalytic arm of IMPACT [46], was used. The patients were offered 28 sessions. The therapy model emphasized general psychodynamic treatment principles with therapist interventions, exploring the young person’s relationships to others, thoughts, feelings and behavior. During therapy sessions, the therapists guided the adolescent through an uncovering process where more of the young person’s unconscious motives and phantasies were revealed. The focus in both groups was on what the patient finds important to talk about; however, it also pointed to themes that the young person might avoid. In the transference work group, the therapists added to the model an encouragement to explore feelings and thoughts about the therapist and the therapy, as well as repetitive patterns of reactions and actions emerging during the sessions in relationship to the therapist. These transference interventions were offered to a moderate level (i.e., 1–3 times per session). In the non-transference work group, these interventions were proscribed.

The therapists were experienced psychologists and psychiatrists especially trained through a one-year course based on the treatment manual [46], with a specific focus on offering psychotherapy with or without transference work. To ensure that the STTP model in each therapy group was delivered, peer supervision groups were offered throughout the study period. Further, sessions were audio-recorded for the purpose of ensuring therapists’ adherence to the treatment arms. The therapists’ use of the specific transference techniques differentiated significantly between the treatment groups.

The level of the transference interventions was measured on a Likert scale 0–4; and was found to be 2.2 (SD 1.47) in the transference work group and 0.52 (SD 0.78) in the non-transference work group (df 52.5, t =−5.5, *p* < 0.0006). The intraclass correlation among two raters (single measure) was 0.89 (CI 95%) [45]. The average number of attended sessions was 18.6 (SD = 8.6) in the transference work group and 18.0 (SD = 10.9) in the non-transference work group.

As a consequence of procedural oversights, six patients had a missing BDI at baseline. These patients still had valid data for SCL-90-R, including the depression subscale. The correlation between the depression subscale and the BDI total score was 0.63, and it was considered sufficient to use this scale to impute the missing BDI at pre-treatment. This was realized by linear regression with SCL-90-R depression scores as the independent variable and the BDI total score as the dependent variable, using the predicted scores in the analyses.

At the one-year follow-up, 22 patients had a missing BDI (32%). In order to rule out that BDI at the one-year follow-up was “missing not at random”, an ANOVA was conducted involving the following four groups: (1) patients in the transference group for whom a BDI at the one-year follow-up was available; (2) patients in the transference group with a missing BDI at the one-year follow-up; (3) patients in the non-transference group for whom a BDI was available; and (4) patients in the non-transference group with a missing BDI. Comparing the number of PD criteria across these groups did not result in any significant differences, neither for the number of cluster B criteria (F = 0.49; *p* = 0.69) nor for the number of cluster C criteria (F = 0.48; *p* = 0.70). Thus, it was assumed that BDI at the one-year follow-up was missing “totally at random”, though a “missing at random” mechanism cannot be excluded. Therefore, no further steps were taken to control for missing values of the covariates in the analyses.

Written consent was obtained from all patients. The patients were included from two areas in Norway: the capital Oslo, and the mixed urban and rural areas in Vestfold. The patients were treated in outpatient clinics. The Central Norway Regional Ethics Health Committee approved the study protocol (https://www.med.uio.no/klinmed/english/research/projects/fest-it/pdf/fest-it_protocol.pdf, accessed on 15 June 2011) (REK: 2011/1424 FEST-IT). The FEST-IT study is registered in ClinicalTrials.gov: NCT01531101.

To measure the level of depression, the patients filled in a self-report scale, the BDI [52], at five time points: pre-treatment; 12 weeks; 20 weeks; post-treatment; and the one-year follow-up.

The analyses for this study were performed by an independent researcher overseen by one senior researcher at the University of Oslo and research services at Oslo University Hospital. One patient had missing PD diagnoses at baseline and was not included in the analyses. A total of 68 of the 69 depressed adolescent patients recruited from outpatient clinics included in the study were included in the intention-to-treat analyses, including data from pre-treatment to post-treatment.

### 2.2. Medication

One patient used antidepressant medication at the beginning of therapy, while one other used it at the end of therapy. One patient was taking antipsychotics throughout the study period. One patient at pre-treatment and four patients at post-treatment were taking sleeping medicine [54].

### 2.3. Data Analysis

The linear mixed model module in SPSS was used to examine moderator effects of TW during the whole study period. This model included five components; an intercept for BDI (β1) to correct for elevated depressive symptoms at baseline; a “time” component (β2) to account for a reduction in BDI scores for the entire sample during treatment; a “time ∗ group” interaction (β3) to correct for differential treatment response across the two treatment groups; “time ∗ number of PD criteria” interaction (β4) to control for potential predictor effects of the number of these PD criteria; and finally, a “time ∗ treatment group ∗ number of PD criteria” interaction (β5) to investigate the moderating effect of the number of PD criteria, i.e., cluster A, B or C criteria. These interactions were included independently of their significance levels. Since there was no significant association between pre-treatment BDI scores and the number of cluster A, B or C PD criteria, this interaction component was not included. The time variable was coded by integers, i.e., 0, 1, 2, 3 and 8, where each integer represents approximately 10 weeks. The final model was as follows:BDIij = β1 + β2 ∗ time + β3 ∗ time ∗ group + β4 ∗ time ∗ PD criteria + β5 ∗ time ∗ group ∗ PD criteria + εij

This five-component model was compared with a simpler model with only three components, i.e., a model in which the latter two interactions were not included. This model is described extensively in the paper of Ulberg et al. [48]. The Akaike information criterion (AIC) and −2 restricted log likelihood (LLH) were used to evaluate model fit. Parametrization was based on maximum likelihood estimation, and models were compared by log likelihood ratio tests. Within-subject association among the vector of repeated response was accounted for by assuming an unstructured covariance pattern. This model had better fit than the mixed effect model in which within-subject association across measurement occasions was accounted for by including random effects at the individual level (see Ulberg et al. [48] for details). Residual plots were inspected for the model with the cluster B PD criteria and did not reveal any aberrations. As outlined in the project description published at ClinicalTrials.gov, the significance level was set at *p* < 0.10.

No significant differences were observed between the transference work group and the comparison group on the pre-treatment variables (Table 2).

## 3. Results

Between February 2012 and September 2017, 100 adolescents were assessed for eligibility. A total of 29 did not meet the criteria for MDD and one declined to participate. A total of 70 were randomized to transference (*N* = 39) or non-transference (*N* = 31) therapy. One patient from the non-transference group withdrew from the study (Table 1).

### 3.1. Moderator Analyses Cluster A Criteria

The results of the moderator analyses are displayed in Table 3. For cluster A criteria, the three-way interaction (“time ∗ treatment group ∗ number of cluster A criteria”) was not significant (t = 1.0; *p* = 0.311), indicating that there was no moderator effect for the cluster A criteria. Thus, an increased level of cluster A pathology at baseline did not seem to have an impact on differential treatment response across to the treatment arms in this study. The two-way interaction was not significant (time ∗ number of cluster A criteria; t = −0.36, *p* = 0.719), suggesting that the number of cluster A criteria was not a predictor of treatment outcome.

### 3.2. Moderator Analyses Cluster B Criteria

For the number of cluster B criteria, the three-way interaction (time ∗ treatment group ∗ number of cluster B criteria ∗) was significant at the α = 0.10 level (t = 2.0, *p* = 0.057), indicating that there was a moderator effect. Closer inspection of the regression parameters (Table 3) revealed that patients with a larger number of cluster B criteria at baseline did significantly better in the transference group than in the non-transference group. A regression coefficient of 0.10 for the three-way interaction (“time ∗ treatment ∗ cluster B criteria”) implied that, for each successive cluster B criterion fulfilled, the non-transference group had a reduction in BDI score that was 0.10 points less than for the transference group, for every 10th week. A regression coefficient for “time” of −2.2 implied that, on average, patients in the transference group had a reduction in BDI of 2.2 points for every time period (10 weeks). The non-transference group, however, had a reduction of 0.9 BDI points every 10th week (−2.2 + 1.3 = −0.9).

Compared with the simpler model, i.e., the model with only three components, LLH went down with 3.8 points (from 1808.7 to 1804.9), whereas AIC did not change (AIC = 1844.9). Improvement of model fit was not significant at the α = 0.10 level.

### 3.3. Moderator Analyses Cluster C Criteria

The three-way interaction (“time ∗ treatment group ∗ number of cluster C criteria”) was not significant (F = 0.27; *p* = 0.60), indicating that there was no moderator effect for cluster C criteria. In other words, patients with higher levels of cluster C pathology at baseline had the same treatment response regardless of which treatment group they belonged to. The two-way interaction was not significant either (time ∗ number of cluster C criteria; F = 0.07, *p* = 0.79), indicating that the number of cluster C criteria could not be considered a predictor of treatment response.

## 4. Discussion

The present study has shown a slight but significant moderating effect of cluster B PD symptoms on the outcome of psychotherapy. Depressed adolescents showed better outcomes when psychodynamic therapy included TW compared to psychodynamic therapy without TW during the whole study period, up to one year post-treatment.

A substantial number of adolescents currently seeking treatment in mental health clinics are suffering from PD symptoms, with as many as 22% clearly exceeding the diagnostic threshold for one or more specific PD diagnoses. Most of these patients display comorbid symptom disorders, with depression being the most frequent one [11,55]

The FEST-IT study previously demonstrated the specific and enduring efficacy of TW on the severity of depressive symptoms in psychoanalytic psychotherapy with depressed adolescents [48,54]. In view of the highly prevalent co-occurrence of adolescent PDs and their precursors, the clinical usefulness of this main finding is further enhanced by exploring the possible moderating effect of PD symptoms on treatment outcome in depressed adolescents.

Transference work has been studied and shown to be effective in psychotherapy with neurotic patients [53]. However, opinions of its usefulness in the treatment of patients with PDs have oscillated between being considered beneficial or even detrimental in the management of both adolescent and adult patients. [36,38]. Studies in this field [37,53] suggest that transference work may help tailor the treatment to the individual patient with a PD.

A core purpose of psychotherapy with young people with depression is the reduction in depressive symptoms. We found a small but significant moderator effect for TW for patients with cluster B personality pathology (assessed by the SIDP-IV), i.e., patients with more severe cluster B personality pathology did somewhat better in treatment when therapists focused on the patient–therapist relationship. However, in our sample, the presence of cluster A and C PD symptoms did not moderate therapy outcomes. Especially in the case of cluster A, this could in part be explained by the small sample size. It could, however, also indicate an actual benefit of TW for patients with an increased amount of cluster B symptoms, but not for those with other types of PD.

Due to the small sample size, it was not possible to break down the data into discrete PD symptom criteria. Still, the results for cluster B disorders, which are highly prevalent in psychiatric outpatient as well as inpatient samples, are of clinical significance, as they indicate that the more personality disordered the individual is, the more he/she stands to gain from TW. The question arises whether this finding could be generalized.

One might speculate whether patients with cluster B PDs are more likely to have deficits in mentalizing, and therefore benefit from a greater focus on directly working with the challenges of maintaining relationships in the therapeutic context. Mentalization refers to the ability to reflect upon, and to understand, one’s own and others’ state of mind [25,54]. As a concept, it is clearly related to the concepts of transference and TW. Severe PDs are generally associated with mentalization deficiencies [37,56]. Patients with cluster B PDs typically have problems recognizing and understanding, let alone accommodating, other people’s, as well as their own, sentiments and emotions. Apart from personal suffering caused by a deficient mentalization ability, this also constitutes a major disadvantage in coping with the challenges of contemporary living, which requires quite a lot of interpersonal interaction, even when performing the simplest of tasks. We cannot function properly in the modern vocational world without being able to communicate, and a prerequisite for developing these skills to a functional level is having an adequate mentalization capacity. Arguably, mentalization, to a certain extent, can be taught as well as learnt. This is a basic concept of the MBT treatment method [31,57] and probably an important factor in all efficacious treatment of severe PDs, regardless of their respective theoretical underpinnings.

The present study is a further corroboration of the importance of talking about relations, especially relations in the therapy room. Our results indicate that addressing mentalization problems that arise in the patient–therapist relationship are more effective interventions than addressing these problems outside the therapy room, at least for patients with severe cluster B personality pathology and MDD. The indication of the present study is that the more seriously disturbed patients seem to be the ones who stand to gain the most from talking about their relations in therapy, i.e., performing TW.

A small sample size should also be considered in light of the complexity of the model. We used a marginal model for longitudinal data in which within-subject association across measurement occasions was accounted for by assuming an unstructured covariance pattern. This model provided a better fit than the competing model, i.e., the random intercept model, but is also more complex by including 20 estimated parameters, whereas the random intercept model included 12 parameters. Thus, the generalizability of the findings might be compromised.

While the results of the present study, with its’ limitations due to the small sample size and the imbalance in gender distribution, do not yield far-reaching conclusions, it could have implications for our therapeutic approach to cluster B PD adolescent patients in general, which are highly prevalent in specialist health services. It is important to note that the young people in the present study suffer from comorbid cluster B PD and MDD. This warrants more research on the effects of expanding TW—talking and negotiating the patient–therapist relationship—into areas of treatment and subgroups of patients where such an approach has been considered not efficacious or even detrimental to therapeutic progress. More RCTs with a focus on moderator analyses are needed in this field. Such studies could reveal which patients would profit most from STPP [46], DBT [29] or MBT [31] and enhance personalized treatment.

## 5. Conclusions

When treated with psychoanalytic psychotherapy for MDD, adolescents with cluster B PD symptoms profited more when encouraged to explore the relationship with the therapist than when this therapist technique was not applied. Adolescents with B PD symptoms profited more from transference work than adolescents without such pathology.

## Figures and Tables

**Table 1 ijerph-19-10952-t001:** Pre-treatment characteristics in 69 adolescents receiving 28 weeks of psychoanalytical psychotherapy with or without transference work [48].

	**Transference Work Group** **(*n* = 39)**		**Non-Transference Work Group** **(*n* = 30)**	
	* **N** *	**%**	* **N** *	**%**
**Gender**				
Female	33	84.6	24	80.0
Male	6	15.4	6	20.0
**Diagnostics**				
Recurrent depression	15	38.5	9	30.0
Prevalence of one or more comorbid diagnoses	18	46.2	16	53.3
	**Mean**	**(SD)**	**Mean**	**(SD)**
**Age**	17.30	(0.7)	17.31	(0.7)
**Personality Diagnostics**				
**PD Criteria as Measured with SIDP-IV**	13.5	(9.0)	12.4	(7.8)

**Table 2 ijerph-19-10952-t002:** Becks Depression Inventory (BDI) measured over time in 69 adolescents receiving short-term psychoanalytic psychotherapy with or without transference work [48].

	Transference Work Group			Non-Transference Work Group		
	N	Mean	(SD)	N	Mean	(SD)
Pre-treatment	35	28.24	(9.7)	28	29.10	(8.4)
12 weeks	28	21.66	(11.7)	19	20.38	(11.7)
20 weeks	21	19.04	(12.4)	15	20.57	(13.2)
Post-treatment	34	15.45	(11.6)	24	17.21	(14.3)
One-year follow-up	26	8.42	(10.9)	20	14.75	(11.9)
*p* < 0.05						

**Table 3 ijerph-19-10952-t003:** Results of the moderator analyses during the whole study period.

	Estimate	90% CI	df	t	F	*p*
**Number of Cluster A Criteria**						
intercept	28.0	26.3 to 29.7	67.8	27.1	735.9	0.000
time	−2.1	−2.6 to −1.6	55.6	−6.6	61.9	0.001
time ∗ treatment	0.40	−0.35 to 1.14	50.7	0.98	0.78	0.381
time ∗ cluster A criteria	−0.05	−0.26 to 0.17	52.5	−0.36	0.40	0.719
time ∗ treatment ∗ cluster A Criteria	0.24	−0.15 to 0.64	50.1	1.0	1.0	0.311
**Number of Cluster B Criteria**						
intercept	28.1	26.3 to 29.8	67.8	27.2	734.7	0.000
time	−2.2	−2.7 to −1.7	55.6	−7.5	40.0	0.000
time ∗ treatment	1.3	0.53 to 2.1	49.3	2.8	8.0	0.007
time ∗ cluster B criteria	−0.02	−0.07 to 0.03	45.9	−0.63	1.4	0.530
time ∗ treatment ∗ cluster B Criteria	0.10	0.01 to 0.19	48.3	2.0	3.8	0.057
**Number of Cluster C Criteria**						
intercept	28.0	26.3 to 29.7	67.8	27.1	733.0	0.000
time	−2.1	−2.8 to −1.4	48.5	−5.4	38.1	0.000
time ∗ treatment	0.36	−0.64 to 1.4	47.3	0.60	0.36	0.553
time ∗ cluster C criteria	−0.01	−0.04 to 0.03	45.0	−0.23	0.14	0.821
time ∗ treatment ∗ cluster C Criteria	0.02	−0.04 to 0.09	45.3	0.64	0.40	0.527

## Data Availability

The data are available from the second author.

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
