# Peer review of "Personality Disorders as a Possible Moderator of the Effects of Relational Interventions in Short-Term Psychoanalytic Psychotherapy with Depressed Adolescents"

_ijerph, 2022, doi:10.3390/ijerph191710952_

Round 1
Reviewer 1 Report
Thank you for the opportunity to review the manuscript “Personality disorders as a possible moderator of the effects of relational interventions in short-term psychoanalytic psychotherapy with depressed adolescents”.
This manuscript addresses an interesting topic, as it investigates the influence of personality disorders (PD), specifically PD clusters, on the effects of the treatment of depressed adolescents. In addition, the therapeutic approach using to treat these adolescents, psychoanalytic psychotherapy, can be considered an unexplored topic. Although the sample is small, I believe it is sufficient to discuss the moderating effect of PD on treatment outcome, as well as the role of the transference work. However, this manuscript presents a set of flaws that compromises the proper evaluation of its scientific quality.
Below are a few noteworthy problems:
1) First, I strongly suggest that authors revised the text, as it presents repetition of words (e.g., lines 85-86), capital letters in the middle of sentences (e.g., line 163), and paragraphs without references (e.g., lines 36-41).
2) I suggest the inclusion of “psychoanalytic psychotherapy” among the keywords, even if “psychodynamic” has to be excluded.
3) Avoid using “you” (e.g., line 400), it is preferable to use “we”.
4) Introduction:
· I suggest the exclusion of headings (depression, personality disorders, etc.), including phrases to link the ideas/paragraphs, if necessary;
· Authors should present more information about depression in adolescence (lines 43-47);
· Reference must be included in the sentence “Adolescence appears also to be a particularly vulnerable period for the development of more complex psychiatric disorders, such as PD”;
· When referring to the % of adolescents that meet diagnostic criteria for borderline personality disorder, the authors could include references to more recent studies (lines 67-69);
· When mentioning a specific study, reference must be included following its mention, e.g., “A study of unselected adolescents [xx]…”;
· The paragraphs on transference seems to be in the wrong order. It would make more sense (and would be connected with the subject of the previous paragraph) to start with "In psychodynamic psychotherapy, however, …” (lines 110-118), following “Transference is a basic concept…” (lines 99-110);
· Kernberg [reference number];
· Hoglend and colleagues [reference number];
· Reference must be included when affirming that “STPP is an evidence-based approach to the treatment of depression in adolescents” (lines 144-145);
· Lines 175-184 should not be in the introduction section, but in the method section;
· Lines 186-191 present the result of a FEST-IT study with depressed adolescents. Would it be a justification for the current study? As there are no references to this statement, it appears that the authors are reporting their own results in the introduction section. If it is indeed the justification for the relevance of the study, the reference must be included, and this paragraph and the sentence below (lines 192-194) must be merged into one paragraph.
5) Method section:
· The description of the study methodology is confusing;
· I suggest changing the order of paragraphs in subsection 2.1:
The paragraph on obtaining written consents and ethics committee approval should be at the end of the subsection, following the lines 243-245 (if it is considered necessary to mention that the analysis for the study was carried out by an independent researcher);
Following the paragraph presenting the sample characteristics and how the diagnoses of MDD and PD were made, it should be presented how the number of 69 adolescents was reached (how many were recruited? how many accepted to participate, how many were treated... and information such as lines 245-248), and how these adolescents were distributed to the therapists (lines 210-212).
Information provided in lines 309-312 must be in the subsection 2.1;
Subsection 2.5 should be merged into subsection 2.1, or following subsection 2.1, as it describes criteria for sample selection;
Then, therapy should be described (lines 213-224; lines 227-231);
The therapists' experience can be mentioned after the therapy description.
· The scales used to measure the level of transference must be described in a specific sub-section, where all assessment instruments are presented;
· All scales must be described including their psychometric properties;
· Since a patient was taking an antipsychotic throughout the study period, would not it be better to exclude him from the study? Or mention/highlight the effects of therapy on him?
· Table 1 should be in the Results section;
· The description of data analysis seems confusing to me. I suggest that the authors revise and make the text clearer and more objective. This subsection must be the last subsection of the method section.
6) Results section:
· The results section should start by presenting the characteristics of the sample (age, gender, pathologies, education, etc.) and, therefore, it seems to make sense that the first table to be presented is table 2;
· Following the description of the sample, the results of the moderating analyzes must be presented, as well as Table 3. Thus, Table 3 must be the second table to be presented in the article;
· Table 1 should be the last table presented in the article, where we can see the results of both groups throughout their treatments. In addition, more information about these results should be presented in the results section.
7) Discussion section:
· The discussion section can be further developed, including the limitations of this study (which is not limited to the sample size);
· “Recent findings by Kernberg and Hoglend suggest…” (line 373). An article published in 2011 is not considered recent. Therefore, it can be said that "studies in this field suggest...";
· Avoid using "you". For example, in line 387, it is better to say "...they indicate that the more personality disordered individual is/patient is, the more she/he stands to gain from TW";
· Authors could more directly propose which studies can contribute to advances in this field, including study design proposals (optional).
8) The conclusion could provide more information about the findings of the present study. But it is ok.
In conclusion, a major revision is needed before this manuscript is processed further.
I hope these comments are a useful guide for you to improve the manuscript.
Sincerely,
Reviewer
Author Response
Dear Reviewer 1,
Thank you for reviewing our paper and for all the very helpful comments.
We think we in the following, have attended pointwise to all input.
Comments and Suggestions for Authors
Thank you for the opportunity to review the manuscript “Personality disorders as a possible moderator of the effects of relational interventions in short-term psychoanalytic psychotherapy with depressed adolescents”.
This manuscript addresses an interesting topic, as it investigates the influence of personality disorders (PD), specifically PD clusters, on the effects of the treatment of depressed adolescents. In addition, the therapeutic approach using to treat these adolescents, psychoanalytic psychotherapy, can be considered an unexplored topic. Although the sample is small, I believe it is sufficient to discuss the moderating effect of PD on treatment outcome, as well as the role of the transference work. However, this manuscript presents a set of flaws that compromises the proper evaluation of its scientific quality.
Below are a few noteworthy problems:
- First, I strongly suggest that authors revised the text, as it presents repetition of words (e.g., lines 85-86), capital letters in the middle of sentences (e.g., line 163), and paragraphs without references (e.g., lines 36-41).
Response: Thank you for making us aware of this. The paper has been revised with focus on spelling and references added.
- I suggest the inclusion of “psychoanalytic psychotherapy” among the keywords, even if “psychodynamic” has to be excluded.
Response: The keyword has been added.
3) Avoid using “you” (e.g., line 400), it is preferable to use “we”.
Response: A change has been made accordingly.
4) Introduction:
- I suggest the exclusion of headings (depression, personality disorders, etc.), including phrases to link the ideas/paragraphs, if necessary;
Response: The headings in the introduction have been deleted.
- Authors should present more information about depression in adolescence (lines 43-47);
Response: More information about adolescent major depressive disorder has been included in paragraph 2.
- Reference must be included in the sentence “Adolescence appears also to be a particularly vulnerable period for the development of more complex psychiatric disorders, such as PD”;
Response: A reference has been included.
- When referring to the % of adolescents that meet diagnostic criteria for borderline personality disorder, the authors could include references to more recent studies (lines 67-69);
Response: A new reference has been included.
- When mentioning a specific study, reference must be included following its mention, e.g., “A study of unselected adolescents [xx]…”;
Response: We agree with the reviewer. The reference which was placed at the end of the paragraph is now included at an earlier point, to make it clearer which paper the text is actually referring to. We have also made similar changes throughout the paper.
- The paragraphs on transference seems to be in the wrong order. It would make more sense (and would be connected with the subject of the previous paragraph) to start with "In psychodynamic psychotherapy, however, …” (lines 110-118), following “Transference is a basic concept…” (lines 99-110);
Response: Thank you for this advice. We have changed the order of the paragraphs accordingly.
- Kernberg [reference number];
- Hoglend and colleagues [reference number];
Response: These references are included at the end of the respective sentences and we have chosen to keep it this way.
- Reference must be included when affirming that “STPP is an evidence-based approach to the treatment of depression in adolescents” (lines 144-145);
Response: Please see previous comment.
- Lines 175-184 should not be in the introduction section, but in the method section;
Response: We fully agree with the reviewer. The paragraph has been moved to the method section.
- Lines 186-191 present the result of a FEST-IT study with depressed adolescents. Would it be a justification for the current study? As there are no references to this statement, it appears that the authors are reporting their own results in the introduction section. If it is indeed the justification for the relevance of the study, the reference must be included, and this paragraph and the sentence below (lines 192-194) must be merged into one paragraph.
Response: The statement has been referenced and the information merged into one single paragraph, as suggested by the reviewer.
5) Method section:
- The description of the study methodology is confusing;
- I suggest changing the order of paragraphs in subsection 2.1:
The paragraph on obtaining written consents and ethics committee approval should be at the end of the subsection, following the lines 243-245 (if it is considered necessary to mention that the analysis for the study was carried out by an independent researcher);
Response: The method section has been reorganized accordingly.
Following the paragraph presenting the sample characteristics and how the diagnoses of MDD and PD were made, it should be presented how the number of 69 adolescents was reached (how many were recruited? how many accepted to participate, how many were treated... and information such as lines 245-248), and how these adolescents were distributed to the therapists (lines 210-212).
Response: The stratification procedure is described in paragraph 6 in the method section in the revised paper. More information on how the number of participants was reached is included at the end of paragraph 2.3.
Information provided in lines 309-312 must be in the subsection 2.1;
Response: We think most of this information is found elsewhere. This information is integrated with the previous change in information about the study and its participants. Please see the previous paragraph.
Subsection 2.5 should be merged into subsection 2.1, or following subsection 2.1, as it describes criteria for sample selection;
Response: The information about the missing data has been merged into paragraph 2.1.
Then, therapy should be described (lines 213-224; lines 227-231);
Response: More information about the therapy has been included.
The therapists' experience can be mentioned after the therapy description.
Response: This is where it now is mentioned.
- The scales used to measure the level of transference must be described in a specific sub-section, where all assessment instruments are presented;
Response: The description of the Likert scale is now in a separate paragraph.
- All scales must be described including their psychometric properties;
Response: The measures are referenced. We tend to think that describing the psychometric properties of each scale would be out of the scope of the present paper and divert attention from the main focus.
- Since a patient was taking an antipsychotic throughout the study period, would not it be better to exclude him from the study? Or mention/highlight the effects of therapy on him?
Response: The study protocol presupposes intention-to-treat-analyses. Therefore, the patient has not been excluded.
- Table 1 should be in the Results section;
Response: Thank you for this suggestion. We have changed accordingly.
- The description of data analysis seems confusing to me. I suggest that the authors revise and make the text clearer and more objective. This subsection must be the last subsection of the method section.
Response: The description of the data analysis is now the last paragraph in the method section. We agree that this is a complicated paragraph. The description is in accordance with the pre-defined statistics plan and the paper of the main findings (Ulberg et al, 2021)
6) Results section:
- The results section should start by presenting the characteristics of the sample (age, gender, pathologies, education, etc.) and, therefore, it seems to make sense that the first table to be presented is table 2;
Response: We agree and have changed accordingly.
- Following the description of the sample, the results of the moderating analyzes must be presented, as well as Table 3. Thus, Table 3 must be the second table to be presented in the article;
Response: Please see next response.
- Table 1 should be the last table presented in the article, where we can see the results of both groups throughout their treatments. In addition, more information about these results should be presented in the results section.
Response: We think that the previous Table 1, now changed to Table 2, is a table presenting descriptive data that should be placed before the moderator analyses. These data are further elaborated in a previous paper from the study. We have chosen not to repeat them here.
7) Discussion section:
- The discussion section can be further developed, including the limitations of this study (which is not limited to the sample size);
Response: More limitations connected with the comorbid sample in the present study has been included.
- “Recent findings by Kernberg and Hoglend suggest…” (line 373). An article published in 2011 is not considered recent. Therefore, it can be said that "studies in this field suggest...";
Response: We have changed accordingly.
- Avoid using "you". For example, in line 387, it is better to say "...they indicate that the more personality disordered individual is/patient is, the more she/he stands to gain from TW";
Response: We have changed accordingly.
- Authors could more directly propose which studies can contribute to advances in this field, including study design proposals (optional).
Response: We have added two sentences with suggestions for future moderator studies.
8) The conclusion could provide more information about the findings of the present study. But it is ok.
Response: Thank you!
In conclusion, a major revision is needed before this manuscript is processed further.
I hope these comments are a useful guide for you to improve the manuscript.
Dear Reviewer 1,
Thank you for reviewing our paper and for all the very helpful comments.
We think we in the following, have attended pointwise to all input.
Comments and Suggestions for Authors
Thank you for the opportunity to review the manuscript “Personality disorders as a possible moderator of the effects of relational interventions in short-term psychoanalytic psychotherapy with depressed adolescents”.
This manuscript addresses an interesting topic, as it investigates the influence of personality disorders (PD), specifically PD clusters, on the effects of the treatment of depressed adolescents. In addition, the therapeutic approach using to treat these adolescents, psychoanalytic psychotherapy, can be considered an unexplored topic. Although the sample is small, I believe it is sufficient to discuss the moderating effect of PD on treatment outcome, as well as the role of the transference work. However, this manuscript presents a set of flaws that compromises the proper evaluation of its scientific quality.
Below are a few noteworthy problems:
- First, I strongly suggest that authors revised the text, as it presents repetition of words (e.g., lines 85-86), capital letters in the middle of sentences (e.g., line 163), and paragraphs without references (e.g., lines 36-41).
Response: Thank you for making us aware of this. The paper has been revised with focus on spelling and references added.
- I suggest the inclusion of “psychoanalytic psychotherapy” among the keywords, even if “psychodynamic” has to be excluded.
Response: The keyword has been added.
3) Avoid using “you” (e.g., line 400), it is preferable to use “we”.
Response: A change has been made accordingly.
4) Introduction:
- I suggest the exclusion of headings (depression, personality disorders, etc.), including phrases to link the ideas/paragraphs, if necessary;
Response: The headings in the introduction have been deleted.
- Authors should present more information about depression in adolescence (lines 43-47);
Response: More information about adolescent major depressive disorder has been included in paragraph 2.
- Reference must be included in the sentence “Adolescence appears also to be a particularly vulnerable period for the development of more complex psychiatric disorders, such as PD”;
Response: A reference has been included.
- When referring to the % of adolescents that meet diagnostic criteria for borderline personality disorder, the authors could include references to more recent studies (lines 67-69);
Response: A new reference has been included.
- When mentioning a specific study, reference must be included following its mention, e.g., “A study of unselected adolescents [xx]…”;
Response: We agree with the reviewer. The reference which was placed at the end of the paragraph is now included at an earlier point, to make it clearer which paper the text is actually referring to. We have also made similar changes throughout the paper.
- The paragraphs on transference seems to be in the wrong order. It would make more sense (and would be connected with the subject of the previous paragraph) to start with "In psychodynamic psychotherapy, however, …” (lines 110-118), following “Transference is a basic concept…” (lines 99-110);
Response: Thank you for this advice. We have changed the order of the paragraphs accordingly.
- Kernberg [reference number];
- Hoglend and colleagues [reference number];
Response: These references are included at the end of the respective sentences and we have chosen to keep it this way.
- Reference must be included when affirming that “STPP is an evidence-based approach to the treatment of depression in adolescents” (lines 144-145);
Response: Please see previous comment.
- Lines 175-184 should not be in the introduction section, but in the method section;
Response: We fully agree with the reviewer. The paragraph has been moved to the method section.
- Lines 186-191 present the result of a FEST-IT study with depressed adolescents. Would it be a justification for the current study? As there are no references to this statement, it appears that the authors are reporting their own results in the introduction section. If it is indeed the justification for the relevance of the study, the reference must be included, and this paragraph and the sentence below (lines 192-194) must be merged into one paragraph.
Response: The statement has been referenced and the information merged into one single paragraph, as suggested by the reviewer.
5) Method section:
- The description of the study methodology is confusing;
- I suggest changing the order of paragraphs in subsection 2.1:
The paragraph on obtaining written consents and ethics committee approval should be at the end of the subsection, following the lines 243-245 (if it is considered necessary to mention that the analysis for the study was carried out by an independent researcher);
Response: The method section has been reorganized accordingly.
Following the paragraph presenting the sample characteristics and how the diagnoses of MDD and PD were made, it should be presented how the number of 69 adolescents was reached (how many were recruited? how many accepted to participate, how many were treated... and information such as lines 245-248), and how these adolescents were distributed to the therapists (lines 210-212).
Response: The stratification procedure is described in paragraph 6 in the method section in the revised paper. More information on how the number of participants was reached is included at the end of paragraph 2.3.
Information provided in lines 309-312 must be in the subsection 2.1;
Response: We think most of this information is found elsewhere. This information is integrated with the previous change in information about the study and its participants. Please see the previous paragraph.
Subsection 2.5 should be merged into subsection 2.1, or following subsection 2.1, as it describes criteria for sample selection;
Response: The information about the missing data has been merged into paragraph 2.1.
Then, therapy should be described (lines 213-224; lines 227-231);
Response: More information about the therapy has been included.
The therapists' experience can be mentioned after the therapy description.
Response: This is where it now is mentioned.
- The scales used to measure the level of transference must be described in a specific sub-section, where all assessment instruments are presented;
Response: The description of the Likert scale is now in a separate paragraph.
- All scales must be described including their psychometric properties;
Response: The measures are referenced. We tend to think that describing the psychometric properties of each scale would be out of the scope of the present paper and divert attention from the main focus.
- Since a patient was taking an antipsychotic throughout the study period, would not it be better to exclude him from the study? Or mention/highlight the effects of therapy on him?
Response: The study protocol presupposes intention-to-treat-analyses. Therefore, the patient has not been excluded.
- Table 1 should be in the Results section;
Response: Thank you for this suggestion. We have changed accordingly.
- The description of data analysis seems confusing to me. I suggest that the authors revise and make the text clearer and more objective. This subsection must be the last subsection of the method section.
Response: The description of the data analysis is now the last paragraph in the method section. We agree that this is a complicated paragraph. The description is in accordance with the pre-defined statistics plan and the paper of the main findings (Ulberg et al, 2021)
6) Results section:
- The results section should start by presenting the characteristics of the sample (age, gender, pathologies, education, etc.) and, therefore, it seems to make sense that the first table to be presented is table 2;
Response: We agree and have changed accordingly.
- Following the description of the sample, the results of the moderating analyzes must be presented, as well as Table 3. Thus, Table 3 must be the second table to be presented in the article;
Response: Please see next response.
- Table 1 should be the last table presented in the article, where we can see the results of both groups throughout their treatments. In addition, more information about these results should be presented in the results section.
Response: We think that the previous Table 1, now changed to Table 2, is a table presenting descriptive data that should be placed before the moderator analyses. These data are further elaborated in a previous paper from the study. We have chosen not to repeat them here.
7) Discussion section:
- The discussion section can be further developed, including the limitations of this study (which is not limited to the sample size);
Response: More limitations connected with the comorbid sample in the present study has been included.
- “Recent findings by Kernberg and Hoglend suggest…” (line 373). An article published in 2011 is not considered recent. Therefore, it can be said that "studies in this field suggest...";
Response: We have changed accordingly.
- Avoid using "you". For example, in line 387, it is better to say "...they indicate that the more personality disordered individual is/patient is, the more she/he stands to gain from TW";
Response: We have changed accordingly.
- Authors could more directly propose which studies can contribute to advances in this field, including study design proposals (optional).
Response: We have added two sentences with suggestions for future moderator studies.
8) The conclusion could provide more information about the findings of the present study. But it is ok.
Response: Thank you!
In conclusion, a major revision is needed before this manuscript is processed further.
I hope these comments are a useful guide for you to improve the manuscript.

Reviewer 2 Report
The study delves into the relationship between depression and personality disorder in adolescents, highlighting Borderline PD as a moderating element. I thank the authors for their research, especially in the psychodynamic field. The work is qualitatively very important because there are few studies that investigate the dynamics of transference. So I think the work deserves to be published, however I would ask the authors to modify the work according to some suggestions.
1) the introduction needs to be broadened, both with regard to research that highlights depressive disorder as the most common occurrence in adolescent girls. also considering the imbalance of the sample, this element should also be discussed more in the discussions.
2) in the theoretical part on depression a sufficiently broad framework is not given to the psychodynamic meaning of depression. Kernberg is mentioned later but much has been written about depression and its meaning by Melanie Klein, a useful element to understand why Borderline disorder in adolescents actually complicates the situation. Furthermore, much literature on these issues uses PDM 2 precisely to enhance the psychodynamic diagnosis.
3) Line 175 and following shows the results of the previous study, but measures and tools are indicated that do not make the meaning of the study understandable. I would suggest telling the results achieved more from a theoretical point of view and then taking up any considerations in the discussions.
4) I would also suggest inserting a paragraph on limits, because although the work is interesting and of quality, there are many limits. Including the number, the imbalance of the gender also the different pharmacological treatments. I understand very well the fatigue of such long studies and with a psychodynamic approach but this element can also be enhanced within the limits themselves.
Author Response
Dear Reviewer 2,
Thank you for reviewing our paper and for all the very helpful comments.
We think we in the following, have attended pointwise to all input:
Comments and Suggestions for Authors
The study delves into the relationship between depression and personality disorder in adolescents, highlighting Borderline PD as a moderating element. I thank the authors for their research, especially in the psychodynamic field. The work is qualitatively very important because there are few studies that investigate the dynamics of transference. So I think the work deserves to be published, however I would ask the authors to modify the work according to some suggestions.
1) the introduction needs to be broadened, both with regard to research that highlights depressive disorder as the most common occurrence in adolescent girls. also considering the imbalance of the sample, this element should also be discussed more in the discussions.
Response: Thank you for revising this paper and for your constructive input. We have added more information about depression in the introduction. The gender imbalance is included in the discussion (limitation)-section.
2) in the theoretical part on depression a sufficiently broad framework is not given to the psychodynamic meaning of depression. Kernberg is mentioned later but much has been written about depression and its meaning by Melanie Klein, a useful element to understand why Borderline disorder in adolescents actually complicates the situation. Furthermore, much literature on these issues uses PDM 2 precisely to enhance the psychodynamic diagnosis.
Response: Additional information about depression has been included in the introduction section. This study has used DSM-diagnoses and a broader elaboration of the psychodynamic understanding of depression would have been very interesting, however, it seems to be out of the scope of this empirical paper.
3) Line 175 and following shows the results of the previous study, but measures and tools are indicated that do not make the meaning of the study understandable. I would suggest telling the results achieved more from a theoretical point of view and then taking up any considerations in the discussions.
Response: Some more qualitative information has been included alongside the description of the results in the outcome measures.
4) I would also suggest inserting a paragraph on limits, because although the work is interesting and of quality, there are many limits. Including the number, the imbalance of the gender also the different pharmacological treatments. I understand very well the fatigue of such long studies and with a psychodynamic approach but this element can also be enhanced within the limits themselves.
Response: Limitations have been discussed at the end of the Discussion section. We have broadened the discussion according to the suggestion of the reviewer.

Reviewer 3 Report
This post-hoc analysis of data taken from the FEST-IT study examined whether personality disorder criteria acted as moderators on treatment outcome regarding depression in adolescents who receive short-term psychoanalytic psychotherapy with transference work versus without. The study provided some interesting initial findings for further research; however, the following should be considered:
1. The title should reflect that personality disorder clusters were studied and not personality disorders.
2. In table 2, the personality disorder criteria by each of the clusters should be given for the two groups and the prevalence of the personality disorders in each of the treatment groups should be provided.
3. In the discussion, for example from line 406, the authors needed to specify that the value of transference work was related to the outcome depression and not related to other outcomes. Thus, it cannot be said that a focus on the patient-therapist relationship will impact the adolescents’ function in other relationships or impact their overall functioning.
Author Response
Dear Reviewer 3,
Thank you for reviewing our paper and for all the very helpful comments.
We think we in the following, have attended pointwise to all input:
Comments and Suggestions for Authors
This post-hoc analysis of data taken from the FEST-IT study examined whether personality disorder criteria acted as moderators on treatment outcome regarding depression in adolescents who receive short-term psychoanalytic psychotherapy with transference work versus without.
Response: Thank you for revising this paper which comprises the analyses of the pre-defined second analysis stated in the protocol. This point is now stated explicitly in the manuscript.
The study provided some interesting initial findings for further research; however, the following should be considered:
- The title should reflect that personality disorder clusters were studied and not personality disorders.
Response: Due to the small sample size, cluster analyses have been performed. Further specification ,i.e., to replace “Personality disorder” in the title with “cluster B PD symptoms” will not provide additional information. Therefore, we tend to prefer the designation “personality disorder” since this more clearly signifies the actual topic of the paper.
- In table 2, the personality disorder criteria by each of the clusters should be given for the two groups and the prevalence of the personality disorders in each of the treatment groups should be provided.
Response: We agree with the reviewer, however, due to the small sample size, we have aggregated the data into PD clusters.
- In the discussion, for example from line 406, the authors needed to specify that the value of transference work was related to the outcome depression and not related to other outcomes. Thus, it cannot be said that a focus on the patient-therapist relationship will impact the adolescents’ function in other relationships or impact their overall functioning.
Response: Thank you for this input. We have now explicitly stated that this study is on comorbid cluster B PD and depression and discussed this in the limitation section.

Round 2
Reviewer 1 Report
Manuscript ID: ijerph-1812980
Title: Personality disorders as a possible moderator of the effects of relational interventions in short-term psychoanalytic psychotherapy with depressed adolescents
Journal: International Journal of Environmental Research and Public Health (IJERPH)
First, I would like to congratulate the authors for the improvement they have made to the manuscript. The text is clearer and more objective. I would also like to mention how pleased I am to see studies on psychoanalytic psychotherapies.
I have just a few notes about the manuscript:
- It seems to me that what is described in lines 231-237 ("the measured primary result...") should be in the results section;
- I do not understand the reason for subsection 2.2;
- The sentence about the descriptive statistics of the BDI being in table 2 is not necessary. In fact, such a sentence is in the methods section and the table is (correctly) presented in the results;
- Subsection 2.3 makes no sense. Everything about the sample selection can be described in subsection 2.1 (Study design and participants).
- References on line 452 must be together, ie "[38,54]".
- Regarding the description of the psychometric properties of all scales, if the journal does not require it, ok. I agree with the authors that there is no need, although it is common for journals to require such information.
Kind regards,
Reviewer 1
Reviewer 2 Report
The manuscript has been improved, I appreciate the new version
